# Optimization and Formulation of Nanostructured and Self-Assembled Caseinate Micelles for Enhanced Cytotoxic Effects of Paclitaxel on Breast Cancer Cells

**DOI:** 10.3390/pharmaceutics12100984

**Published:** 2020-10-18

**Authors:** Farah Rehan, Nafees Ahemad, Rowshan Ara Islam, Manish Gupta, Siew Hua Gan, Ezharul Hoque Chowdhury

**Affiliations:** 1School of Pharmacy, Monash University Malaysia, Jalan Lagoon Selatan, Bandar Sunway 47500, Petaling Jaya, Selangor, Malaysia; farah.rehan@monash.edu (F.R.); nafees.ahemad@monash.edu (N.A.); manish.gupta@monash.edu (M.G.); gan.siewhua@monash.edu (S.H.G.); 2Tropical Medicine and Biology Multidisciplinary Platform, Monash University, Jalan Lagoon Selatan, Bandar Sunway 47500, Petaling Jaya, Selangor, Malaysia; 3Global Asia in the 21st century Research Platform, Monash University, Jalan Lagoon Selatan, Bandar Sunway 47500, Petaling Jaya, Selangor, Malaysia; 4Jeffrey Cheah School of Medicine and Health Sciences, Monash University, Jalan Lagoon Selatan, Bandar Sunway 47500, Petaling Jaya, Selangor, Malaysia; Rowshan.Islam@monash.edu; 5School of Pharmaceutical and Population Health Informatics, DIT University, Mussoorie-Diversion Road, Dehradun, Uttarakhand-248009, India

**Keywords:** natural polymers, sodium caseinate, paclitaxel, colloidal stability, cytotoxicity, nanomicelles, breast cancer, tumor regression

## Abstract

Background: Paclitaxel (PTX) is a widely used anti-cancer drug for treating various types of solid malignant tumors including breast, ovarian and lung cancers. However, PTX has a low therapeutic response and is linked with acquired resistance, as well as a high incidence of adverse events, such as allergic reactions, neurotoxicity and myelosuppression. The situation is compounded when its complex chemical structure contributes towards hydrophobicity, shortening its circulation time in blood, causing off-target effects and limiting its therapeutic activity against cancer cells. Formulating a smart nano-carrier may overcome the solubility and toxicity issues of the drug and enable its more selective delivery to the cancerous cells. Among the nano-carriers, natural polymers are of great importance due to their excellent biodegradability, non-toxicity and good accessibility. The aim of the present research is to develop self-assembled sodium caseinate nanomicelles (NaCNs) with PTX loaded into the hydrophobic core of NaCNs for effective uptake of the drug in cancer cells and its subsequent intracellular release. Methods: The PTX-loaded micelle was characterized with high-performance liquid chromatography (HPLC), Fourier Transform Infrared Spectra (FTIR), High Resolution-Transmission Electron Microscope (HR-TEM), Field Emission Scanning Electron Microscope (FESEM) and Energy Dispersive X-Ray (EDX). Following treatment with PTX-loaded NaCNs, cell viability, cellular uptake and morphological changes were analyzed using MCF-7 and MDA-MB 231 human breast cancer cell lines. Results: We found that PTX-loaded NaCNs efficiently released PTX in an acidic tumor environment, while showing an enhanced cytotoxicity, cellular uptake and in-vivo anti-tumor efficacy in a mouse model of breast cancer when compared to free drug and blank micelles. Additionally, the nanomicelles also presented improved colloidal stability for three months at 4 °C and −20 °C and when placed at a temperature of 37 °C. Conclusions: We conclude that the newly developed NaCNs is a promising carrier of PTX to enhance tumor accumulation of the drug while addressing its toxicity issues as well.

## 1. Introduction

Despite substantial advances in therapeutic strategies, breast cancer remains the second leading cause of cancer-related deaths in women [1]. In fact, the World Health Organization (WHO) reported that breast cancer is the most common type of cancer, impacting over 1.5 million women every year. According to the American Cancer Society, 40,610 women were likely to die as a result of breast cancer in 2017 [1,2], indicating a high disease burden.

PTX is a systemic chemotherapeutic agent used to treat advanced and metastatic breast cancer due to its microtubule-stabilizing property [3,4]. Its response rate against breast cancer ranges from 30–70% [1]. The therapeutic efficacy tends to be limited due to energy-dependent efflux of the drug following its internalization into cancer cells via passive diffusion, leading to the development of multidrug resistance (MDR) [5,6].

The ATP-driven efflux is facilitated by various transmembrane transporters of the ATP-binding cassette (ABC) superfamily. Since P-glycoprotein (Pgp; ABCB1; MDR1) is one of the leading transporters and expressed in various breast cancer cell lines [6,7], an effective drug delivery system with a unique mechanism of drug transport across the cancer cells might overcome the MDR [8]. Apart from the acquired resistance, PTX also possesses many severe side effects, such as peripheral sensory neuropathy. Moreover, it has a poor aqueous solubility (0.7–30 μg/mL) due to its high lipophilicity [9,10]. Researchers have attempted various excipients to enable parenteral administration of PTX, with cremophor E.L. (polyoxyethylated castor oil derivative) and dehydrated ethanol (50:50 *v*/*v*) successfully used to date for intravenous infusion of taxol [9]. Nevertheless, the formulation faces various drawbacks, including anaphylaxis and other severe hypersensitivity reactions [10].

Nano-scaled drug carriers for cancer treatment have shown various benefits when compared to conventional formulations, including (1) reduced side effects due to less systemic toxicity, (2) increased half-life and (3) tumor targetability of drugs. To date, various PTX-bound nano-sized carriers including lipid-based nano-delivery system, polymeric nanoparticles, nanocrystals, carbon nanotubes and cyclodextrin nanoparticles have been developed [11] as a promising approach to treat cancers effectively.

Nano-drug carriers are assumed to bypass by the membrane-associated, Pgp-mediated drug efflux by facilitating drug delivery via endocytosis rather than through passive diffusion, which could help in overcoming drug resistance [7]. Furthermore, nano-carriers can promote the passive targeting of the drugs utilizing leaky vasculature structure and impaired lymphatic drainage system of solid tumors via enhanced permeability and retention (EPR) effect [7,12]. Commercially approved nanoparticulate formulations including Abraxane^®^, (a human serum albumin-bound PTX) [13] and Lipusu^®^ (PTX-loaded liposomes) [14] are considered as efficient Cremophor EL-free PTX formulations. They also tend to confer fewer side-effects as compared to Taxol. However, due to the poor colloidal stability of Abraxane in the blood [15], as well as minimal improvement of other clinically approved PTX-bound nanoformulations (in terms of response rate and therapeutic efficacy), an improved innovation for the delivery of PTX is required [11].

Polymeric micelles (PMs) confer several advantages over other nano-sized drug delivery carriers, for example, having a smaller size (50–200 nm), thus allowing it to extravasate into cancer tissues via EPR effect. The easy accessibility enhances the anti-cancer effects while providing a minimal toxicity to healthy tissues [16,17,18]. Additionally, PMs are block-co-polymers with a hydrophobic core stabilized by a hydrophilic shell when self-assembled in an aqueous environment. The hydrophilic shell provides a steric stability to micelles that can avoid the rapid uptake by the reticuloendothelial system (RES) and assist in prolongation of its circulation in the body [16]. 

To date, the FDA-approved Genexol-PM or [methoxypoly(ethylene glycol)-b-poly(lactic acid)] (mPEG-PLA micelle) [19] shows promising results in clinical practice, although some hypersensitivity issues remained. Therefore, for clinical implementation of PMs, further development regarding drug loading, intracellular drug release and tumor accumulation is required. Moreover, there is an urgent need to avoid the high cost involved in micelles preparation under a good manufacturing practice (GMP) condition due to multiple modifications or attachment of targeted ligands [16].

A nano-drug delivery system based on natural material is highly desirable to overcome the various problems related to PTX formulation [20]. Interestingly, casein is one of the significant components of milk, transports vital nutrients, such as protein, calcium and phosphate, from a mother to a neonate as a natural nano delivery system, while meeting the energy and growth requirements of mammalian neonates [6,21]. Therefore, as a natural and inexpensive protein [22], casein nanospheres possess various excellent properties, such as biocompatibility, amphiphilicity, rapid dispersion in the aqueous media, a self-assembling nature and feasibility for large scale production [23,24]. Casein molecules are made up of the 4 cas phophoproteins (αS1-Cas, αS2-Cas, β-Cas, and κ-Cas) to form a diblock ampiphilic copolymers. This natural structure of casein allow the micelles to self-associate into core-shell nanocomposite, where the hydrophobic block aggregate to form the inner core while hydrophilic blocks form the outer nanocomposite shell [25].

Moreover, a casein micelle enables intravenous injection of poorly soluble anti-cancer drugs by incorporating them inside its hydrophobic core [26]. Previously, enhanced cytotoxic effects were observed when PTX [27] was incorporated inside the hydrophobic core of the β-casein in gastric carcinoma. Similar results were also obtained with other hydrophobic drugs, including resveratrol [22] and flutamide [28], when encapsulated into casein, mainly via the hydrophobic interactions [21]. In the present research, we aimed to develop nanostructured NaCNs as a natural micellar system for more efficient and selective delivery of PTX to breast cancer cells both in vitro and in vivo. 

## 2. Materials and Methods

### 2.1. Chemical Reagents

Sodium caseinate from bovine milk, PTX and acetone were purchased from the Sigma-Aldrich, Eschenstrasse, Taufkirchen, Germany. Dulbecco’s Modified Eagle Medium (DMEM) was bought from Nacalai Tesque Inc, Nakagyo-Ku, Kyoto, Japan. Fetal Bovine Serum (FBS) and penicillin-streptomycin (P/S) was purchased from Gibco, Life Technologies, Loughborough, UK, while 3-(4,5-dimethyl thiazolyl-2)-2,5-diphenyltetrazolium bromide (MTT) was obtained from Merck, Petaling Jaya, Selangor, Malaysia. Acetonitrile, methanol and isopropyl alcohol (all HPLC grades) were obtained from Chemmart Asia Sdn Bhd, Puchong, Selangor, Malaysia. Bicinchoninic Acid (BCA) protein kit was purchased from Sigma-Aldrich, Eschenstrasse, Taufkirchen, Germany, MCF-7 and MDA-MB 231 cell lines were originally from ATCC (Manassas, VA, USA).

### 2.2. Methods

#### 2.2.1. Synthesis of Sodium Caseinate Nanomicelles (NaCNs)

An optimized synthesis methodology was employed for the generation of NaCNs. Briefly, different amounts of sodium caseinate (ranging from 0.125 to 10.00 mg) were mixed in ultra-pure water (1 mL) and then vortexed to obtain a clear solution. Finally, the samples were placed on a nutating mixer under a gentle mixing mode for 1 to 8 h, to optimize the formation of micelles in the aqueous media.

#### 2.2.2. Optimisation of Blank NaCNs

First, different parameters (concentration and mixing times) were optimized for the preparation of blank NaCNs. The samples were then further optimized through Particle size analysis, protein content determination, turbidity measurement, optical microscopy and FESEM analysis. 

#### 2.2.3. Particle Size Measurement

Particle size and Zeta potential were measured to determine the optimum concentration of the sodium caseinate required to make micelles at particular mixing time. Different amounts of sodium caseinate (0.125, 0.25, 0.50, 1, 2, 4, 6, 8 and 10 mg) was added to 1 mL of distilled water in an Eppendorf tube. Nine samples were prepared, as stated above and were vortexed before being placed on the nutating mixer. Sodium caseinate nanomicelles (NaCNs) were obtained after 1, 2, 4 and 8 h of mixing time. After ten times dilution (optimized) of NaCNs samples, the particle size and zeta potential were measured using a Malvern Nano Zetasizer (Malvern, Worcestershire, UK). All the measurements were performed at 25 °C and in triplicates. They were recorded as mean ± SD.

#### 2.2.4. Protein Analysis

To determine the amount of protein involved in the preparation of micelles, protein analysis was performed using a BCA protein kit. Different amounts of sodium caseinate (0.125, 0.25, 0.50, 1, 2, 4, 6, 8 and 10 mg) were added to 1 mL of distilled water in Eppendorf tubes and placed on a nutating mixer. After 1, 2, 4 and 8 h of mixing time, 100 µL from the samples was carefully obtained and diluted with 200 µL of distilled water. Subsequently, diluted samples were placed in a centrifuge and run for 20 min at 15,000 rpm. Then, 25 µL of supernatant was removed from the tube and pipetted out into 96 well plates. Approximately 200 µL of BCA reagent was added into each sample for protein analysis. Before measuring the absorbance, the samples were incubated at 37 °C for 30 min following the addition of BCA reagent and were shaken for 30 s. After measuring the absorbance of samples, the concentration of free protein was then estimated from the equation of the standard calibration curve (drawn for different concentrations of the sodium caseinate). All the measurements were performed at 25 °C and in triplicates with the results reported as mean ± SD. The percentage of the total proteins associated with NaCNs was calculated [29,30] using the following formula:
(1)$$\text{Protein content}\left(\%\right)=\frac{\text{Proteins in pellet}}{\text{Total proteins initially added}}\times 100$$


#### 2.2.5. Turbidity Measurement and Microscopic Images

Subsequently, turbidity of NaCNs samples prepared with different concentrations and at different mixing duration was measured to analyze the possible effect of concentration and time on the formation of micelles, at 400 nm using a UV-Vis Spectrophotometer. Turbidity measurement is an indirect method to determine the concentration and time-dependent changes in optical density due to micellar growth in a supersaturated solution [31]. For turbidity analysis, different amounts of sodium caseinate (0.125, 0.25, 0.50, 1, 2, 4, 6, 8 and 10 mg) were added to 1 mL of distilled water in Eppendorf tubes. Nine samples were prepared, as stated above and placed on the nutating mixer. After a 1 h of mixing, the samples absorbance was measured at 400 nm (optimized) [32]. Similar procedure was followed after 2, 4 and 8 h of incubation. All measurements were performed at 25 °C in triplicates with the results reported as mean (±SD).

To observe the aggregation patterns among micelles prepared with different concentrations, optical images were taken using an optical microscope (Inverted trinocular microscope Nikon TS100, Minato-Ko, Tokyo, Japan) at a scale bar of 100 μm and a magnification of 10×. Following micelles formation, the samples were transferred to a six well plates, and then microscopic images were taken.

#### 2.2.6. Field Emission Scanning Electron Microscope (FESEM) Analysis

Finally, morphological analysis was also conducted through FESEM images to validate the data obtained by spectrophotometry and optical microscopy. NaCNs samples of different concentrations (1, 2, 4, 6, 8 and 10 mg/mL) for FESEM analysis were prepared. After one h of mixing, each sample (10 µL) was taken on the glass cover and placed until get dried. The samples were then placed on the sample holder coated with carbon tape, followed by a platinum sputtering was carried at 30 mA sputter current for 40 s with 2.30 tooling factor.

#### 2.2.7. Determination of the Critical Micelle Concentration (CMC)

To determine the CMC of NaCN micelles, pyrene fluorescent method (steady-state fluorescence method) was used [33,34]. Pyrene is commonly used as a fluorescent probe to define the environmental polarity of casein micelles [33]. Different concentrations of NaCN from 0.2 to 2 mg/mL were prepared. Then pyrene was added at a concentration of 1 × 10^−6^ M to the casein of varying concentrations and vortexed for 30 s. Fluorescence measurements were carried out through Perkin Elmer LS55 Fluorescence Spectrometer, with the excitation wavelength at 295 nm, and the emission spectrum scanned from 300 to 500 nm. During the experiment, the excitation and emission slits were fixed at 3.0 and 2.5 nm, respectively, and the scan rate was maintained at 240 nm/min. Fluorescence intensity in the monomer band at peak 1 (I_1_ = 373 nm) and peak 3 (I_m_ = 383 nm) and in the excimer band at 450 nm as a function of the casein concentration was recorded to determine the CMC.

#### 2.2.8. Synthesis of PTX-Loaded NaCNs

Self-assembled NaCNs were designed and optimized to incorporate PTX into the hydrophobic core of the micellar system. NaCNs loaded with PTX (PTX-NaCNs) were prepared as that for the blank with some slight modification. Briefly, PTX (10 µM) was allowed to be entrapped inside the micelles by dissolving the drug solution together with sodium caseinate (1 mg) in an aqueous media (1 mL). The resulting solution was vortexed and placed on a nutating mixer with gentle mixing (24 rpm) at 25 °C for one hour which would allow PTX to be physically incorporated inside the micellar core via hydrophobic interactions. Negative control (free PTX + water) was also similarly prepared, albeit without the addition of sodium caseinate powder.

#### 2.2.9. Physicochemical Characterization of PTX-NaCNs

Particle size distribution and zeta potential of both empty and PTX-loaded NaCNs samples were measured by using a dynamic light scattering (DLS) technique [35] utilising a Malvern nano zeta sizer (Malvern, Worcestershire, UK) that was coupled with a Zeta sizer software (version 6.20, Malvern, Worcestershire, UK). The diameter and zeta potential of nanomicelles were measured after diluting the samples with ultrapure water (1:10) to avoid multiple scattering of casein micelles [36]. All experiments were performed at 25 ± 0.1 °C at a viscosity of 0.8872 cP. The measurements were taken three times, and the mean results were reported. The percentage encapsulation efficacy (E.E.) of PTX-NaCNs was measured through HPLC. Subsequently, the characteristic band of PTX which appeared in the loaded micelles was validated by an HPLC calibration study where an excellent linearity (0.9975) was seen for concentrations ranging between 0 to 50 µM (Appendix A). 

The samples were prepared as described above. This was followed by a centrifugation step at 15,000 rpm (4 °C) for 20 min (Eppendorf Centrifuge 5424 R, Sigma Aldrich, Taufkirchen, Germany). Subsequently, the supernatant was carefully collected for HPLC analysis for the presence of free PTX using an Agilent system with an L.C. software coupled to a diode array detector (DAD) detector. The system was equipped with a C18 column (4.6 × 150 mm, 5 μm) that was maintained at 30 °C. Acetonitrile/water (55:45) was used as a mobile phase with a flow rate of 1.5 mL/min. An injection volume of 4 µL was used throughout the analysis. Detection of PTX was conducted at 227 nm. Finally, E.E. and drug loading (D.L.) (*w*/*w*) were calculated by using the formulae below [Equations (2) and (3)] as previously reported [22].
(2)$$\%\;E.E.=\frac{\text{Mass of the drug in micelles}}{\text{Total drug mass used in formulation}}\times 100$$
(3)$$\%\;\mathrm{DL}.=\frac{\text{Mass of the drug in micelles}}{\text{Mass of micelles recovered}}\times 100$$


To investigate the shape of micelles, the samples were examined using FESEM [37] for morphological analysis. Prior to analysis, the formulations were prepared as described above. Then, 10 μL from each formulation was placed onto the glass cover before being air-dried at room temperature. The samples were mounted on a sample holder that was coated with some carbon tape, followed by platinum sputtering (15 s). Subsequently, both the negative control and PTX-NaCNs samples were observed under the FESEM machine (Hitachi/SU8010, Tokyo, Japan) at 5.0 kV. For EDX analysis, observation was conducted at 15.0 kV. For HR-TEM analysis [38], the samples were prepared by suspending the copper grid (300 mesh size) in the nanomicelles suspension for 30 s. Then, the samples were air-dried at room temperature and were further analyzed (without staining) through the HR-TEM machine (FEI tecnai G2 20S-TWIN, Amsterdam, Netherlands) at 200 kV.

The infrared (I.R.) spectra of sodium caseinate, free PTX, blank and PTX-NaCNs as well as their respective physical mixtures, were recorded using a Varian FTIR along with a Varian Resolution Pro 640 software (version 5.1, Agilent, Santa Clara, CA, USA) over the range of 500–4000/cm at an ambient temperature [39].

#### 2.2.10. In Vitro Drug Release Profile

PTX-NaCNs were prepared as described in Section 2.2.8 and subsequently placed into a tube-O-dialyzer that was earlier subjected to a soaking process for several minutes in ultra-pure water. The molecular cut-off (MWCO) was set at 15K Da. Then the solution was suspended inside a beaker containing 40 mL of phosphate buffer saline (PBS) (7.4/5.0 pH). The entire system was maintained at 37 ± 2 °C and subjected to a continuous magnetic stirring (100 rpm) for 24 h. At the same time, free PTX solution was dialyzed using the same dissolution media. Samples (100 uL) were taken from the tube-O-dialyzer and were mixed with acetone (50:50) to allow the release of drug from the nanomicelles. Subsequently, the samples were centrifuged at 15,000 rpm for 20 min at 4 °C. Then the supernatant was carefully collected for HPLC analysis of PTX using the same methodology as used for the encapsulation study.

#### 2.2.11. Systemic and Physical Stability of NaCNs

NaCNs were prepared as described in Section 2.2.8. They were then placed in an incubator at 37 °C in order to mimic the body’s physiological conditions. The aliquots from the sample formulation were taken at specific time intervals and analyzed for particle size and zeta potential. The turbidity of the samples was also checked at 400 nm using a UV-Vis spectrophotometer. For physical stability, accelerated stability studies of nanomicelles were conducted over three months in which the formulations were separately stored at three different temperatures (4 ± 2 °C, −20 ± 2 °C and 25 ± 2 °C) in a desiccator. Subsequently, the particle size, zeta potential and sample turbidity were measured monthly.

#### 2.2.12. In Vitro Cell Viability and Cytotoxicity Studies

The cell viability and cytotoxicity studies were conducted in MCF-7 and MDA-MB-231 cell lines using an MTT assay [22,31]. Briefly, both cell types were cultured with DMEM media having 10% FBS and 1% P/S antibiotic in a 25 cm^2^ flask. The flask was subsequently placed in a humidified incubator at 37 °C and 5% carbon dioxide (CO_2_). Then, the cells were sub-cultured in a 75 cm^2^ flask and were trypsinized once they achieved an exponential growth phase. 

After counting for the number of cells using a hemocytometer, the cell suspension (100 µL) containing 50,000 cells/mL was seeded in each well of a 96-well plate and was placed in a humidified incubator with 5% CO_2_ at 37 °C. After 24 h, the medium was replaced with a fresh medium (100 µL) containing different PTX concentrations either as a free PTX solution or encapsulated in NaCNs and the blank NaCNs along with a control (only media) for another 24 and 48 h. Then, PTX- NaCNs as well as free PTX were diluted in complete DMEM, leading to the final PTX concentrations (0.125–5.00 µM). After 24 and 48 h, the MTT stock solution (20 µL) (5 mg/mL in PBS) was added into each well followed by incubation for another 4 h at 37 °C in the dark. Subsequently, DMSO (100 µL) was added into each well to dissolve the formazan crystals. The cells were agitated in a built-in plate shaker for 10 s. Finally, the absorbance was measured using a microplate reader (Glomax Explorer GM3500, Promega Corporation Australia, Alexandria, NSW, Australia) at 560 nm against 600 nm (reference wavelength). 

The percentage cell viability (% CV) was calculated by using a formula:
(4)$$\%\;\mathrm{Cell}\;\mathrm{Viability}\;\left(\mathrm{CV}\right)=\frac{\text{Absorbance of treated cells}}{\text{Absorbance of control}}$$
while the percentage cell cytotoxicity of free PTX, PTX-NaCNs and blank NaCNs was calculated by using formulae:
(5)% Cytoxicity of Free Drug=[CV (control)−CV(Free Drug)]×100
(6)% Cytoxicity of PTX−NaCNs=[CV (control)−CV(PTX−NaCNs)]×100
(7)% Cytoxicity of Blank−NaCNs=[CV (control)−CV(Blank−NaCNs)]×100


The concentration which caused a 50% inhibition (IC_50_) for the free and micelle-bound PTX was also calculated using a GraphPad Prism (version 8.0, GraphPad Software, San Diego, CA, USA). The data were calculated as a mean of triplicate readings and plotted as a percentage of data from control cultures (in which no drug was added). Standard deviation (S.D.) was calculated for the mean values.

#### 2.2.13. Light Microscopy

Apart from measuring cell viability through the MTT assay, light microscopy images were also taken by using an inverted microscope (Inverted trinocular microscope Nikon TS100) as previously suggested by Ibiyeye et al. [40]. Briefly, blank NaCNS, PTX-NaCNs and free PTX at two concentrations (1 and 5 µM) were prepared. The cells were seeded on a 24-well plate at the concentration of 50,000 cells per well. Following 24 h incubations in a humidified incubator 5% CO_2_ at 37 °C, the cells were treated with blank NaCNs, PTX-NaCNs and free PTX following media removal and were analyzed under an inverted microscope after 24 and 48 h.

#### 2.2.14. Cellular Uptake

The cells were seeded in 24-well plates with a density of the 50,000 cells per well and were incubated for 24 h [41]. After 24 h incubation, the media was removed and was replaced with the free PTX and PTX-NaCNs with final concentrations of 5 and 10 µM. The control (media) was prepared by treating cells with a DMEM complete media. After 4 h of treatment, the supernatant (100 µL) was taken carefully from each well for detection of free PTX which were not taken up by cells. Cellular uptake was quantitatively measured by HPLC following the same methodology as previously described in the encapsulation study.

#### 2.2.15. In Vivo Study

##### Animals

In vivo experiments were performed using Female Balb/c mice aged 7–8 weeks (19 ± 2 g). The animals were acquired from the School of Medicine and Health Science animal facility, Monash University. The experiments were conducted after being approved by the Monash Animal Ethics Committee (Project ID: 2020-19843-39399, 31/1/2020). All animals were housed in a stainless-steel mesh cages under standard conditions of temperature and humidity, having free access to water ad libitum throughout the study, while the light conditions were set at 12:12 h (light and dark ratio).

##### Induction of Breast Tumor in Mice

Murine breast cancer cells (derived 4T1 cells) were cultured in DMEM complete media containing 1% P/S antibiotic and 10% FBS in a 25 cm^2^ flask and later placed in a humidified incubator (at 37 °C and 5% CO_2_). Cells were further subcultured in a 75-cm^2^ flask and were further trypsinized once the exponential growth phase is achieved. After counting of cells through a hemocytometer, the cells were re-suspended in DMEM at 10^6^ cells/mL and later suspended in PBS to obtain a concentration of 10^5^ cells/100 µL. 

The cells were then injected into the left side of the mammary gland at 1 × 10^5^ cells/100 µL subcutaneously using a 27 G needle, now deemed as “day 1”. The mice which developed tumors were observed at least three times weekly until a palpable tumor nodule appeared. Once the tumor was palpable, daily monitoring was conducted. Approximately one to two weeks following inoculation, the mice were randomly assigned to each treatment group (*n* = 5) when the tumor reached a mean volume of 13.53 ± 2.51 mm^3^. The tumor size (both the length and the width) was monitored at regular intervals until the end of the study, with the help of a digital Vernier calliper. 

The following formula was used to measure tumor volume:
(8)$$\mathrm{Tumour}\;\mathrm{volume}\;\left({\mathrm{mm}}^{3}\right)=\frac{1}{2}\left(\mathrm{length}\times {\mathrm{width}}^{2}\right)$$


##### In Vivo Anti-Tumor Efficacy 

In order to evaluate the anti-tumor efficacy, the animals were divided into four groups (*n* = 5 each): (1) negative control (saline) (2) blank NaCNs (3) free PTX and (4) PTX-NaCNs when the tumor size reached approximately 13.53 ± 2.51 mm^3^. Tumor-induced mice were administered with free PTX solution, PTX-NaCNs, while the control groups received saline and blank NaCNs. The mice were administered with a single I.V. injection of free PTX and PTX-NaCNs (1 mg/kg/day) on day 13 following tumor induction. This was done by using a 29 G needle via the tail vein, while the second dose was given two days after the first dose. The mice’s body weight was also measured in every three days. At the end of the experiment (on day 28), mice were exposed to 100% CO_2_ inside the CO_2_ chamber to render them unconscious. Subsequently, the animals were killed by cervical dislocation before collection of their vital organs, such as brain, spleen, lungs, liver, kidney and heart, as well as any visible tumors. The excised tumor and different organs of the mice were washed in cold PBS. Later, the organ weights were also determined after the treatment course. All measurements were presented as mean ± S.D.

#### 2.2.16. Statistical Analysis

Statistical analysis of the in vitro cellular work as well as the in vivo data was conducted using a GraphPad prism (version 8.0). Analysis of Variance (ANOVA) and Turkey’s Multiple Comparison tests were used for the pair-wise comparison analysis. Values at *p* < 0.05 were considered as statistically significant, with a 95% confidence interval (CI).

## 3. Results and Discussion

### 3.1. Optimizing Synthesis of NaCNs 

The main objective of the present research was to optimize the synthesis of casein micelles by investigating the impact of different parameters, such as the concentration and the mixing time on the physicochemical properties of micelles. Different sodium caseinate concentrations (0.125–10.000 mg/mL) with variable mixing times (1–8 h) were investigated with the aim of forming smaller sized micelles with lesser aggregation impact. Mixing samples for 1 and 8 h was found to be better than 2 and 4 h (*p* < 0.05) through particle size analysis (Figure 1 and Figure 2), indicating that mixing time and incubation time could be inversely related, where the higher mixing time may be preventing aggregation while the shorter incubation time was minimizing it. Thus, at 1 hour, smaller sizes were observed in samples while and at 2 and 4 h, bigger particles were found, whereas at 8 h longer time mixing might limit the aggregation process induced by the prolonged incubation time, also causing the formation of smaller particles. Moreover, due to different electrostatic interactions between casein molecules, affecting casein stability, we obtained diverse zeta potential values in the samples prepared with different casein concentrations and mixing time. 

To further optimize the mixing time and concentration of sodium caseinate required for the formation of nanomicelles, turbidity measurement was conducted at 400 nm. It was observed that turbidity values were almost proportional to sodium caseinate concentration. The increasing trend in turbidity was also seen when the mixing time for micelle formation was increased (Figure 3a). It is plausible that the said phenomenon is contributed by protein-protein interactions which trigger sample aggregation at higher concentrations of sodium caseinate as well as at a longer duration of exposure during the mixing process.

Additionally, the growth kinetics of the micellar formation also increased with higher sodium caseinate concentration, resulting in the formation of aggregates. Thus, the samples prepared following 1 hour mixing time conferred lesser micellar aggregation and were therefore deemed to be optimum. Determination of protein content further confirmed that the maximum yield of protein-based micelle was obtained in the samples prepared within one hour of mixing where more than 99% protein was successfully transformed into micelles (Figure 3c). A higher protein content in the micellar formation (almost 100%) also convened the optimal loading of the drug into the micelles [29].

For systemic validation of the results obtained from the turbidity and protein content determination, optical images of NaCNs were further captured. The optical images were taken immediately following the formation of NaCNs in order to visualize NaCNs aggregation when formulated using a higher concentration of sodium caseinate after mixing for one hour. Additionally, a homogenous batch of micelles with uniformly distributed micelles of smaller size (Figure 4) was observed in the sample prepared at 1 mg/mL. In contrast, visible aggregates of micelles were observed in samples formulated with a higher concentration (2–10 mg/mL) of sodium caseinate. The gradual increase in particle aggregation with increased concentration was indicative of high energies created by the micelles when a higher concentration of caseinate was used [42]. Thus, the data generated through the optical microscope further validated our hypothesis that 1 mg/mL of sodium caseinate is the suitable concentration to prepare micelle within 1 h of mixing time.

FESEM morphological analysis further supported the described phenomenon since it helps in understanding the nanostructure of samples [43]. The sample prepared with one hour of mixing in the presence of caseinate (1 mg/mL) yielded micelles of smaller size with visible spherical morphology and lesser aggregation (as shown in Figure 5). Thus more detailed nanostructured analysis through FESEM, further proved that 1 mg/mL concentration of sodium caseinate is deemed suitable to prepare micelle in one h of mixing time. Once the micellar formation with lesser aggregation is confirmed to have been yielded, other tests including turbidity measurement, protein content determination and optical images of the micelles were conducted using lower caseinate concentrations (0.125, 0.250 and 0.500 mg/mL) and a one-hour mixing. Nevertheless, a gradual decrease in turbidity was noticed in caseinate concentration lower than 1 mg/mL, indicating that using lower concentration did not allow sufficient micellar formation. The results were further validated via protein content determination and optical image analysis. Finally, a 1 mg/mL caseinate was selected since it was deemed to be suitable for NaCNs formation for an hour mixing time.

### 3.2. Determination of CMC

Pyrene fluorescent probe was utilized to determine the CMC of NaCN micelles due to its hydrophobic nature which would allow it to interact with hydrophobic domains of the micelles, enabling to localize environmental polarity changes. Figure 6A exhibits the fluorescent intensity in the first (I_1_) and the third (I_m_) vibrant bands as well as the broad excimer band at about 450 nm (I_e_) where the excited monomer encountered with the ground state pyrene to form an excimer [33], while the emission intensity ratio between the first peak to the third (I_1_/I_m_) demonstrates the micro-environmental polarity. Figure 6B shows the abrupt change in the I_1_/I_m_ as the casein concentration reaches between 0.8 to 1 mg/mL, indicating CMC of the casein around 0.9 mg/mL, since the intensity ratio of the monomers increases sharply near the CMC, until it reaches the twice of the initial values. Both the excimer band intensity (Ie) and the ratio of Ie to the monomer (I_1_) for pyrene (I_e_/I_1_) increase strongly near the CMC (0.9 mg/mL) and decrease slowly at higher concentration of the casein (Figure 6B) due to the distribution of the bound pyrene among the increasing concentration of micelles. Self-assembled nanomicelles having low CMC value can avoid the segregation after their exposure to highly diluted conditions [44].

### 3.3. Characterization of PTX-Loaded NaCNs 

Studies have shown that nanoformulations with particle size greater than 200 nm generally do not extravasate into the tumors [45] and therefore are not effective in delivering drugs into fast-growing tumors. In contrast, nanoparticles with smaller particle size (2–10 nm) can easily be removed from the body via the kidneys [46] as well as interstitial and lymphatic fenestration [47] which tend to minimize the desired tumor accumulation and localization [46]. These events can eventually affect the EPR effect, which play an essential role in drug internalization into tumor. Hence, while designing nanoparticles, factors such as the size and surface charge of the particles are important considerations.

In our research, the mean NaCN size was approximately 332.7 nm. However, loading of PTX into the micelles reduced its size further to 198.2 nm (Table 1), which may occur as a result of inhibition in the self-association process of the micelles. Our findings are in line with the study conducted by Shapira et al. [48], where PTX binding was shown to influence the primary micellar aggregation found in the blank formulation, resulting in smaller-size nanomicelles. Additionally, broad size distribution occurred due to casein reassembled nature but as the figure (Figure 7A) shows more than 90% of micelle intensity lies in the 200 nm particle size range, indicating the suitable particle size of our micelle. As long as the particles remain within the size range suitable for systemic administration and able to escape the leaky tumor vessels to reach the tumor, and prevent renal clearance and uptake by reticuloendothelial system, they could be used as smart nanocarriers, previously many studies in our lab [41,49,50] and also some other studies [51,52,53] supported that particles even though heterogeneous in size could be a fantastic nanocarrier system for efficient and tumor-targeted delivery of anti-cancer drugs and genetic materials.

Nevertheless, encapsulation of PTX presented no apparent influence on the zeta potential of NaCNs (Figure 7A) which remained −1.87 mV (empty NaCNs) or at −1.86 mV (PTX-NaCNs), thus suggesting that no electrostatic interaction was involved in PTX-NaCNs formation. 

The surfaces of both NaCNs and PTX-NaCNs showed negative zeta potentials due to the presence of ƙ-casein around casein micelles, which contains negatively charged ester phosphate and carboxylate groups [23]. PTX-NaCNs (Figure 7B) also exhibited a lower polydispersity index (0.469) as compared to NaCNs (0.664), thus suggesting a lower tendency of PTX-loaded micelles to aggregate [54] and the analysis was considered good. Hence, PTX-NaCNs possess tremendous potential in entering tumor cells through passive targeting of cancer cells based on an EPR effect, which can eventually improve the intra-tumoral penetration, cellular uptake and tumor growth inhibitory effect [55] of PTX-NaCNs. PTX showed an E.E. of 50.9% (*v*/*v*) with a D.L. of 3.569% (*w*/*w*), confirming the high payload capacity of NaCNs.

FESEM results (Figure 8A) revealed the spherical morphology and the uniform distribution of the micelles in PTX-NaCNs as compared to the negative control (free PTX + water) (Figure 8B), indicating that the micelle might self-assemble into a spherical shape after encapsulating PTX into its hydrophobic core with the result of a reduction in its overall size. TEM microphotograph (Figure 8 C) also revealed the spherical morphology and re-assembling nature of casein in the PTX-NaCNs sample. It is plausible that the particle size discrepancies yielded from the different analytical techniques are also contributed by the difference in their respective operating principles. Nevertheless, EDX analysis (Appendix A) confirmed the presence of the uniformly distributed sodium (Na), oxygen (O), calcium (Ca) and carbon (C) in the samples to be the main components of the NaCNs. 

FTIR spectroscopy was performed in order to confirm the formation of caseinate-based micelles and their interactions with PTX molecules (Figure 9). The spectrum for sodium caseinate showed the presence of some characteristic peaks at 3399.65, 1794.75 and 1589.7 cm^−1^, originating from N-H stretching and amide-bending vibrations, while that of PTX showed an influential band at 2996.46 cm^−1^ due to C-H and O-H stretching bonds and at 1700.109 cm^−1^ due to a strong C-O stretching bond. The characteristic peak in the regions from 1600–1400 cm^−1^ showed the aromatic ring stretching frequency, the presence of aromatic C-C stretching bond and the bending of C-H bonds. On the other hand, the unique peak from 1300–1100 cm^−1^ showed the C-H bending, C-O stretching and C-N stretching bonds. These characteristics peak were also observed in previous studies [56,57]. Similar characteristics peaks for PTX also appeared in the PTX-NaCNs and the physical mixture of PTX and NaCNs, indicating successful loading of PTX in the NaCNs without any change in the functional groups. No additional peak was observed in the formulation, thus confirming the lack of chemical interaction between NaCNs and PTX.

### 3.4. In Vitro Drug Release Profile

PTX release profile from NaCNs was evaluated in PBS (pH 7.4) in order to mimic the typical physiological environment. Additionally, it was also assessed at pH 5.0 in order to mimic the acidic environment of the tumor and to understand how different pHs could affect the release profile of PTX from a formulation. It was observed that PTX release at both pHs (Figure 10A) was slightly different where the initial burst release of PTX was reduced at a lower pH (pH 5.0) and became more controlled as compared to the higher pH (pH 7.4). The current finding is similar to that reported by Sinaga et al. [36], who stated that the casein micelle size was reversible in the pH range of 6–7. In contrast, the particle size increased above pH 7.5 and decreased at a lower pH (pH 5.0–5.5). 

With an increase in pH, casein molecules lose their structures due to stronger electrostatic repulsion forces, which leads to an increase in the size of casein micelles. On the other hand, at a lower pH (5.5) close to the isoelectric point (pI) of casein, the hydrophobic interaction decreases the electrostatic repulsion between casein molecules. Thus, casein molecules get closer to one another and tend to form a more compact structure with reduced casein size [33], resulting in a more controlled release of PTX.

Therefore, the slight difference in the release kinetics of NaCNs at both pHs (5.0 or 7.4) may be an added advantage for the delivery system since there was no significant difference between the acidic extra- and intracellular tumor environments [58]. PTX-NaCNs followed a biphasic and sustained release profile, by first showing a burst release of approximately 30% in the first 2 h, followed by a more sustained and controlled release for up to 24 h. A similar release profile of casein protein was also reported by El-far et al. [22]. More than 50% of drug was released at pH 7.4, whereas approximately 40% of drug release was seen in 24 h at pH 5.0, indicating that the drug might disperse uniformly in micelles and could leave by diffusion. Previous studies [59,60,61] also demonstrated similar results where micelles tended to reveal a burst release, followed by a slow release of the drug. In contrast, a burst release of free PTX occurred at both pHs, where 50% was released in the first hour at pH 5.0 while at least 40% was released in the subsequent 2 h at pH 7.4 (Figure 10A). The sustained release of PTX from NaCNs further supported the notion that NaCNs could limit the exposure of PTX in the blood and to healthy tissues as well, which might enhance the targeting ability of the micelles to cancerous cells while limiting systemic toxicity [15].

### 3.5. Systemic and Physical Stability of NaCNs

Since polymeric micelles often face low colloidal stability, their dissociation in systemic circulation indicates another stability challenge [26]. To determine the stability of micelles in the systemic circulation, NaCNs were prepared as previously mentioned and placed in the incubator at 37 °C, thus mimicking the body temperature to ensure that the drug is delivered from loaded micelles effectively inside tumor while maintaining its stability at the normal physiological temperature. 

Turbidity tended to increase with an increase in incubation time (Figure 11A), which indicated that the number of particles becomes higher with an increase in incubation time. Moreover, the mean particle size and polydispersity index analyses (Figure 11B) demonstrate that the mean size and polydispersity index values decreased with increase in the incubation time as compared to the initial size of micelles, at physiological temperature (37 °C). The decrease in PDI is indicative of decreased aggregation of the micelles, leading to a higher number of particle formation as indicated by the turbidity measurements. Thus, the decrease in size and PDI values of micelles [Figure 11A(i) and A(ii)] at body temperature would promote better bioavailability of the loaded drug inside the body. The effect of dilution following systemic administration of micelles and their subsequent interactions with serum proteins might further decrease their size and PDI, thus increasing half-life of the drug. On the other hand, no change in the zeta potential shedded light on the stability of the micelles at 37 °C [Figure 11A(iii)].

For physical stability analysis, the samples were taken out at a monthly interval (up to 3 months) and suitably diluted to determine their mean sizes and zeta potentials. Micelles stored at 25 °C showed a drastic increase in particle size (from 546.2 ± 76.76 to 1673 ± 77.72) in 30 days, followed by the slight decrease to 1530 ± 83.81 in 60 days (Figure 12A). At higher temperature, the increased thermal energy might induce the aggregation of micelles by accelerating the interactions between them; hence at 25 °C the particle size increased. However, micelles tended to be more stable both at 4 °C and −20 °C, since the particle size did not increase after 60 days. Additionally, the zeta potential of samples remained negative while the PDI values (Figure 12B,C) showed no significant change throughout the storage time, suggesting that the steric repulsion which is caused by ĸ-casein hairs located on the outer side of micelles contributed towards micellar stability [62].

The need of the drug compatibility inside the formulations was deemed necessary. However, here, we optimized the micelle to observe the micelle stability without drug loading over a period of time.

### 3.6. In Vitro Cell Viability and Cytotoxicity

The anti-tumor efficacy of PTX-NaCNs was investigated against MCF 7 and MDA-MB 231 cell lines and was compared with free PTX. MCF-7 cell model has been extensively studied by various researchers to define estrogen-stimulated growth in tumor due to the estrogen receptor (E.R.) expression [41,63]. On the other hand, MDA-MB 231 (E.R. negative) is the most aggressive breast cancer cell line and known to be resistant to various anti-cancer drugs [63]. 

Most of the formulations demonstrated excellent results in MCF-7 but found to be resistant in MDA-MB 231 cells due to their resistant nature. PTX-NaCNs (Figure 13) showed considerable cytotoxicity (approximately 90%) against both MCF 7 and MDA-MB 231 cell lines when compared with free PTX, using concentrations ranging from 0.125 to 5 µM. Moreover, IC_50_ was also calculated from the cell viability data. After 24 and 48 h of incubation, the IC_50_ of free PTX was 2.168 and 1.976 μM, respectively, against MCF 7 cells. On the other hand, PTX-NaCNs showed superior cytotoxic effect as demonstrated by the reduction of IC_50_ to 1.505 μM and 1.016 μM, respectively, which is almost 1.44 (24 h) and 1.79 (48 h) fold lower than that of free PTX.

When compared against MDA-MB 231 cells, free PTX showed a higher IC_50_ (2.142 μM at 24 h and 1.789 μM at 48 h), as compared to MCF 7 cells due to the resistant nature of MDA-MB 231. On the other hand, the IC_50_ of PTX-NaCNs was significantly lower at 24 h (1.796 μM) and 48 h (0.914 μM) as compared to free PTX. PTX-NaCNs tend to be potent and showed immediate affect after 24 h; however, due to nontoxicity of the nanomicelles some of the cells might further proliferate, showing slightly higher cell viability at 48 h as depicted in Figure 14B,D.

Overall, our results were more significant as compared to that for NK105 (synthetic polymeric micelles) prepared by Hamaguchi et al. [64] which indicated no significant difference between IC50 values of NK105 and PTX against breast cancer cell lines and showed equivalent cytotoxicity.

In contrast, PTX-NaCNs showed an enhanced cytotoxicity due to its smaller size, which helps in adsorbing sufficient PTX and subsequently, enables a more effective cellular internalization via endocytosis. The higher toxicity could also be by virtue of overcoming MDR [6,24], which resulted in an increased accumulation of PTX-NaCNs inside the drug-resistant cell lines [65] as compared to free PTX [66]. Moreover, the enhanced cytotoxicity of PTX-NaCNs against both types of breast cancer cell lines could be due to the ability of casein micelles to penetrate cancer cells [67], thus resulting in an increased accumulation of PTX into the cancer cells [22,68]. Additionally, compared to three controls (untreated, blank micelles and free PTX), PTX-loaded micelles showed considerably higher cytotoxicity due to efficient binding of PTX to the micelles, and the results were further correlated with our in vivo studies, where PTX-loaded micelles significantly lowered the size of tumor as compared to free PTX and the blank micelles.

Blank NaCNs on the other hand, demonstrated excellent biocompatibility [24] with more than 75% cell viability (Figure 13) in almost all concentrations of sodium caseinate used to prepare NaCNs [69].

### 3.7. Light Microscopy

Light microscopy was employed to visually investigate cell death by analysing the cell morphology (cell rounding, shrinkage and detachment) [40,70]. PTX-NaCNs showed a disrupted morphology probably due to the entrance of micelles into the cells (Figure 14). After 24 h, the cells showed some rounded morphology and were detached from the flask. Following 48 h of exposure, a more widespread cellular fragmentation along with fewer cell adherence was observed when compared with free PTX, emphasizing that drug incorporated in casein micelles possesses extraordinary cell-penetrating ability as compared to the free PTX. It is plausible that free PTX which enters the cells via passive diffusion is highly dependent on drug concentration gradient across the plasma membrane, while PTX-loaded NaCNs are internalized through endocytosis, releasing its “cargo” (PTX) far from the cell membrane [71]. Blank micelles were not as effective when compared with the control (media) and displayed a flattened morphology even after 48 h, which was similarly reported by Sahu et al. [72]. Overall, the negligible toxicity and excellent biocompatibility of the micelles is indicative of their good potential as a safe delivery system for in vivo administration of anti-cancer drugs [66]. 

### 3.8. Cellular Uptake 

Free PTX showed a lower cellular uptake in MCF-7 cells, 13.23 ± 0.21% (5 µM) and 53.18 ± 1.008% (*p* < 0.0001) (10 µM) after 4 h treatment (Figure 10B). However, PTX-NaCNs yielded a notably higher cellular uptake at 42.29 ± 2.43% (5 µM) and 66.40 ± 1.37% (10 µM) (*p* < 0.0001) after 4 h of treatment. As explained above, the difference in the efficiency of cellular uptake of free and micelle-bound PTX could be attributed to the nature of the routes of cellular internalization. Endocytosis of PTX-loaded NaCNs is probably more efficient as compared to a passive diffusion of free PTX, which is dependent on drug concentration gradient across the cell membrane. The passive diffusion might be related to the drug resistance triggered by membrane-anchored efflux pumps [73]. In contrast, micelles are known to be internalized by cells via energy-dependent adsorptive endocytosis. At lower concentrations, the micelles showed approximately 30% more cellular uptake than free PTX, which is well correlated with our cell viability studies. 

### 3.9. In Vivo Anti-Tumor Effect of PTX-NaCNs

The anti-tumor efficacy of the optimized PTX-NaCNs was evaluated in a murine breast cancer model (Figure 15C). Breast cancer-bearing mice were randomly divided into four groups; (1) positive control (tumor-induced mice receiving saline) (2) drug-free NaCNs (3) free PTX solution and (4) PTX-NaCNs. 4T1 cell lines were used to induce tumor in mice. Although 4T1 cells are quite different from the MCF-7 cells; as triple negative breast cancer cells (TNBC), they have similar resistance nature as MDA-MB 231 where PTX-NaCNs showed promising cytotoxic results.

Tumor-induced mice were later given with a free PTX solution and PTX-NaCNs at 1 mg/kg/day for two alternative days using a 29 G needle via their tail veins.

There was no significant reduction in tumor volume or continued growth of tumor throughout the treatment span in the group treated with blank NaCNs, indicating that NaCNs is non-toxic towards tumor cells (Figure 15A). However, following the administration of the two doses, the marked reduction in tumor volume was recorded in PTX-NaCNs-treated group on day 17 (7.52 ± 8.03 mm^3^), followed by a slow tumor growth when compared with free PTX (56.96 mm^3^ vs. 302.70 mm^3,^ respectively) during the study completion. Thus, it was concluded that the antineoplastic activity of PTX is significantly enhanced when being loaded inside the micelles (*p* < 0.001).

The successful tumor regression seen with PTX-NaCNs is attributed to the more controlled and biphasic release of PTX through NaCNs with a burst release of about 30% in the first 2 h, followed by the controlled release up to 24 h. When compared with free PTX injected group, PTX is thought to be slowly released from the hydrophobic core of micelles, exerting a more continued tumor growth suppressive effect [22]. By preventing the premature release of an anti-cancer drug in the blood, NaCNs probably enhanced the half-life of the drug in the blood and thus maximized its concentration in the tumor, resulting in a more enhanced therapeutic effect. Secondly, the natural assembling property of sodium caseinate into nanomicelles facilitated the passive delivery and retention of the drug into tumor cells via an EPR effect [23] with enhanced anti-tumor efficacy. 

No apparent change in the body weight was observed among the groups treated with PTX-NaCNs and NaCNs (Figure 16B), signifying the non-toxic nature of the delivery system. However, there was a slight decrease in the body weight of mice in the free PTX-treated group, which may be due to the high cytotoxicity of the PTX in its free form. Furthermore, no significant change or abnormal behaviour in the mice was visible throughout the entire treatment period.

To further determine the biosafety and toxicity of the nanomicelles, the vital organs including the heart, brain, liver, lung, kidney and spleen of all the treated groups were removed after the treatment period and were individually weighed (Figure 15B). Free PTX-treated groups presented a significant (*p* < 0.001) increase in the spleen and liver size as compared to PTX-NaCNs- treated group, indicating that NaCNs might ameliorate the off-target adverse effects of anti-cancer drugs following systemic administration [22]. A bioavailability study in the future would be helpful for better understanding of the mechanism related to the tumor response to the nanoformulation of PTX.

## 4. Conclusions

We have successfully optimized casein micelles to encapsulate PTX and delivered it, free of cremphor E.L. and ethanol. The optimized PTX-NaCNs showed the particle size of 198 ± 9.81 nm. The drugs were successfully loaded into the micelles with an E.E. of 50.98 ± 4.00%. PTX-NaCNs showed a significantly enhanced cellular uptake and in vitro cytotoxicity effects against MCF-7 and MDA-MB 231 breast cancer cells as compared to free PTX. NaCNs also released PTX in a controlled manner at both pHs 5.0 and 7.4 and showed improved colloidal stability for three months at 4 °C and −20 °C. The primary features of the developed formulations include a more enhanced solubility and cytotoxicity of PTX after being loaded into micelles. The enhanced anti-cancer therapeutic efficacy of PTX-NaCNs was confirmed in a murine breast cancer model. Therefore, NaCNs should be considered as a promising nanocarrier for efficient delivery of a hydrophobic drug like PTX to cancer cells in vivo.

## Figures and Tables

**Figure 1 pharmaceutics-12-00984-f001:**
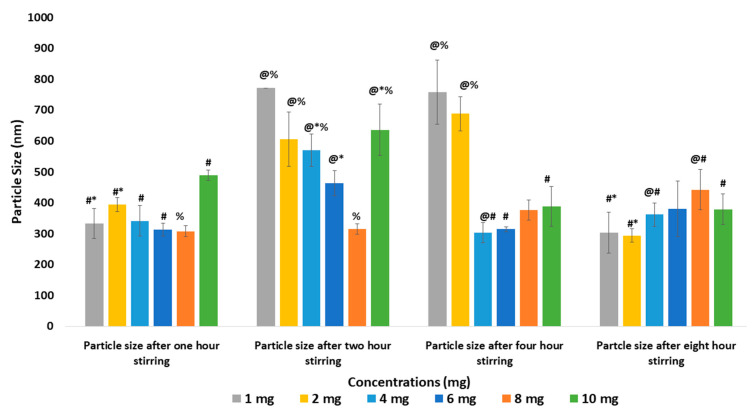
Comparative analysis of particle size of NaCNs prepared with 1, 2, 4, 6, 8 and 10 mg/mL of sodium caseinate and at 1, 2, 4 and 8 h of mixing time. Data are shown as mean ± S.D. where *n* = 3 and @ *p* < 0.05 vs. 1 h, # *p* < 0.05 vs. 2 h, * *p* < 0.05 vs. 4 h, % *p* < 0.05 vs. 8 h mixing time.

**Figure 2 pharmaceutics-12-00984-f002:**
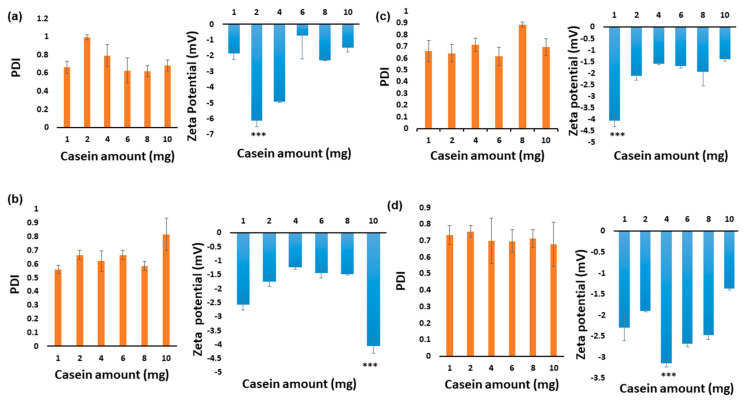
Comparative analysis of polydispersity index (PDI) and size of NaCNs prepared with 1, 2, 4, 6, 8 and 10 mg/mL of sodium caseinate and at (**a**) one hour (**b**) two hours (**c**) four hours (**d**) eight hours of mixing time, values were found very significant at *p* < 0.001 (***).

**Figure 3 pharmaceutics-12-00984-f003:**
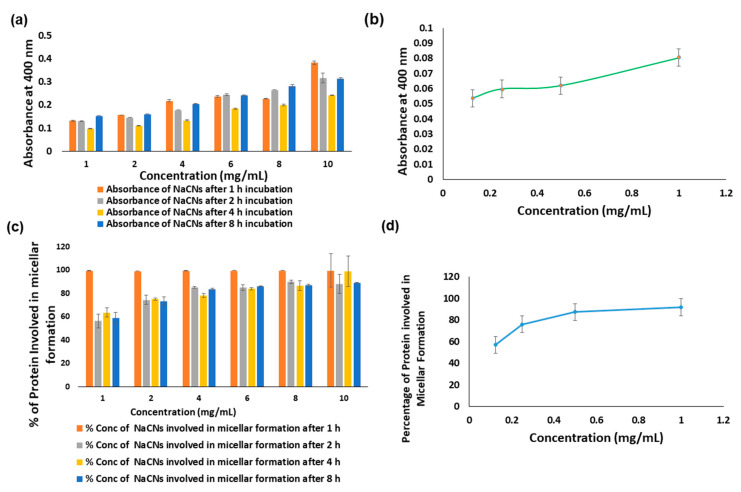
Graphical analysis of (**a**) turbidity measurement of NaCNs prepared with different concentrations of sodium caseinate (1, 2, 4, 6, 8 and 10 mg/mL) and at different mixing times (1, 2, 4 and 8 h); (**b**) turbidity measurement of NaCNs prepared with lower concentrations of sodium caseinate (1, 0.5, 0.25 and 0.125 mg/mL) and at one hour mixing time; (**c**) estimation of protein contents (using BCA protein kit) in NaCNs prepared with different concentrations of sodium caseinate (1, 2, 4, 6, 8 and 10 mg/mL) and at different mixing times; (**d**) estimation of protein contents in NaCNs prepared with lower concentrations of sodium caseinate (1, 0.5, 0.25 and 0.125 mg/mL) and at different mixing times.

**Figure 4 pharmaceutics-12-00984-f004:**
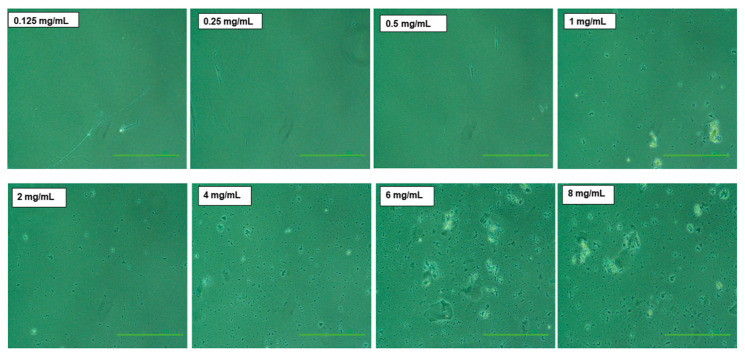
Microscopic images of the aggregates of NaCNs (formulated at different concentrations of sodium caseinate) taken after one hour of mixing time. Scale bar: 100 μm and a magnification: 10×.

**Figure 5 pharmaceutics-12-00984-f005:**
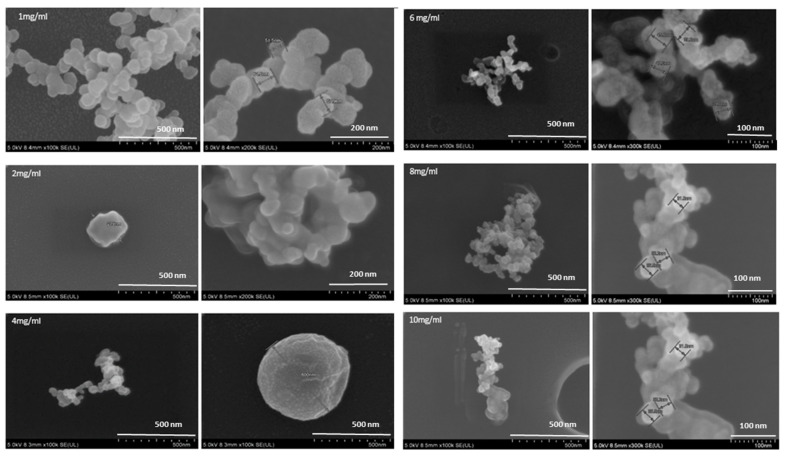
FESEM analysis of the blank NaCNs formulated with 1, 2, 4, 6, 8 and 10 mg/mL concentrations taken after one hour of mixing.

**Figure 6 pharmaceutics-12-00984-f006:**
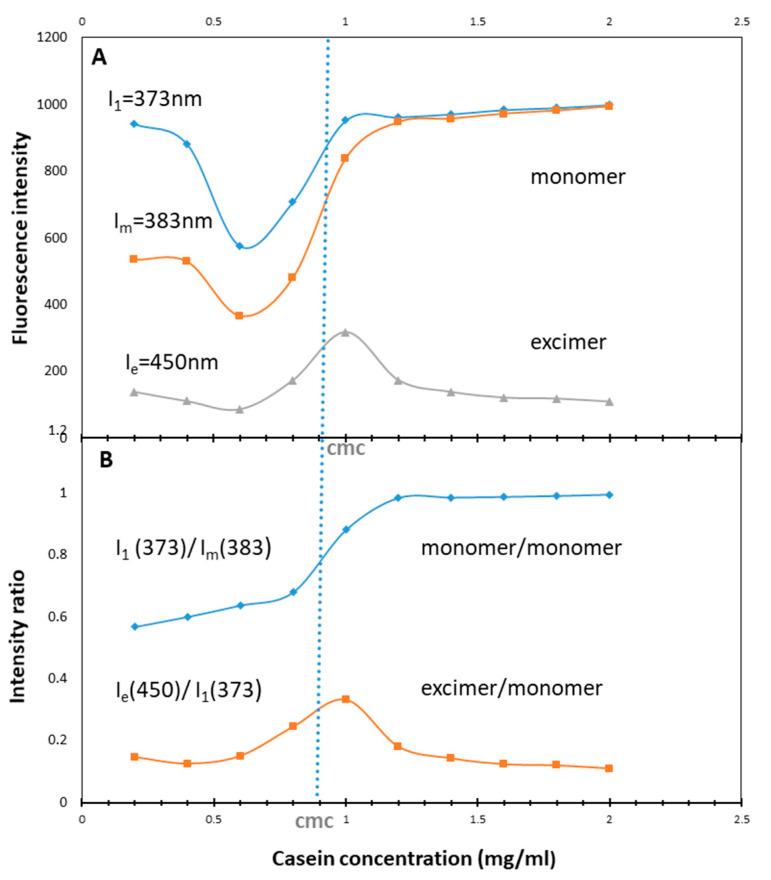
Fluorescence emission of pyrene in aqueous solutions of different concentrations of the casein, ranging from 0.2–2.0 mg/mL (**A**) Fluorescence intensity in the monomer band, at peak I (373 nm, circles) and peak III (383 nm, squares) and in the excimer band at 450 nm, (triangles) as function of the casein concentration. (**B**) Intensity ratios: I_1_/I_m_ (circles) and I_e_/I_1_ (triangles). The dashed blue vertical line designates the cmc of the caseinate micelles.

**Figure 7 pharmaceutics-12-00984-f007:**
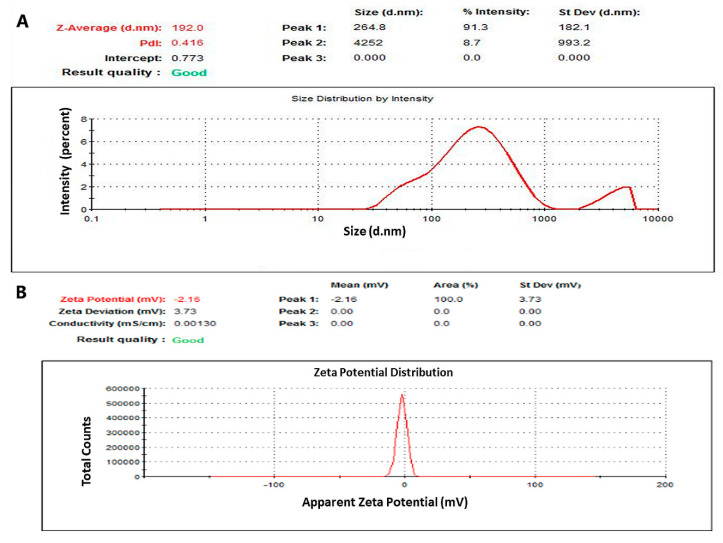
Graphical representation of particle size distribution produced via a DLS technique for determination of (**A**) particle size distribution based on the intensity of PTX-NaCNs and (**B**) zeta potential distribution of PTX-NaCNs.

**Figure 8 pharmaceutics-12-00984-f008:**
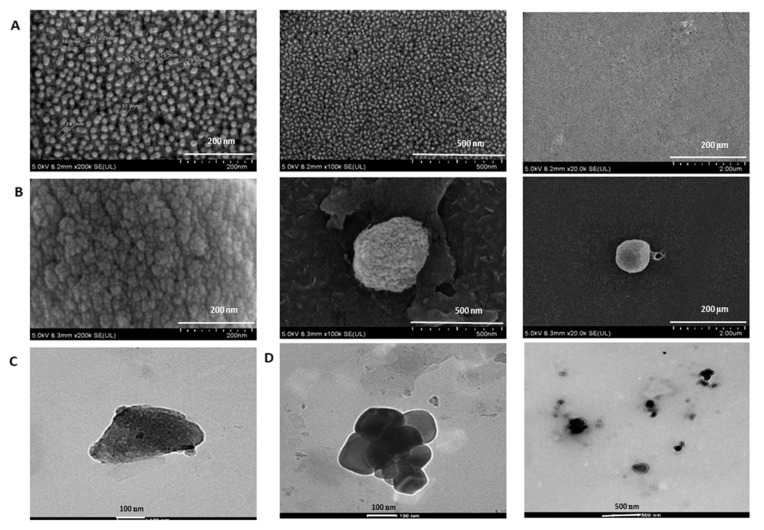
Morphological analysis of NaCNs through FESEM: (**A**) PTX-NaCNs (**B**) Negative Control (PTX + water). Scale bar: 200 nm, 500 nm and 2 µm. Samples (10 µL) were transferred to the glass cover and placed at room temperature until getting dried for FESEM analysis. Morphological analysis of NaCNs through HR-TEM: (**C**) Blank NaCNs (**D**) PTX-NaCNs. (Scale bars: 100 and 500 nm). Copper grid is suspended in the samples for 30 s, air-dried at room temperature and then analyzed through HR-TEM machine.

**Figure 9 pharmaceutics-12-00984-f009:**
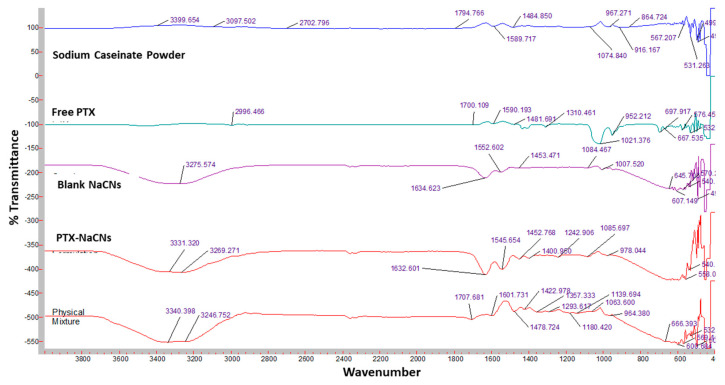
FTIR spectra of Sodium Caseinate powder, PTX, Blank NaCNs, PTX-NaCNs and physical mixture.

**Figure 10 pharmaceutics-12-00984-f010:**
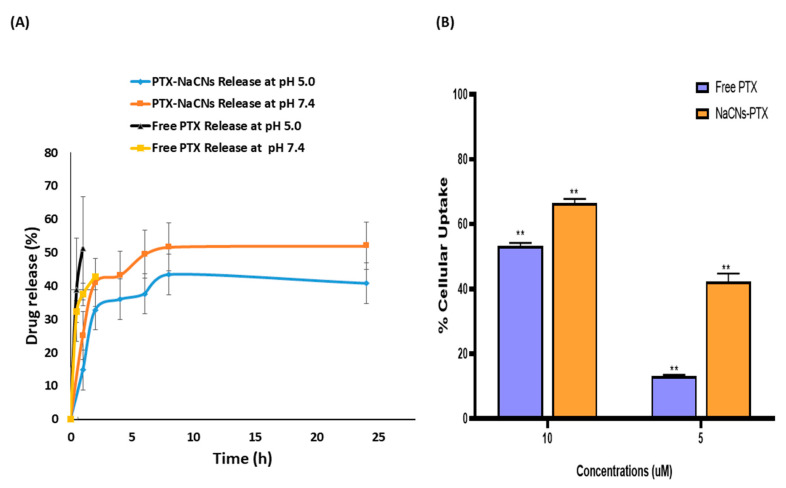
(**A**) In vitro release profile of PTX-NaCNs at pH 5.0 and 7.4, as seen by using a Tube-O-dialyzer (**B**) Cellular uptake of free PTX and PTX-NaCNs by MCF-7 cells at 5 and 10 µM following 4 h of treatment. The values were extremely significant (**) (*p* < 0.0001) at CI of 95% vs. free PTX treatment.

**Figure 11 pharmaceutics-12-00984-f011:**
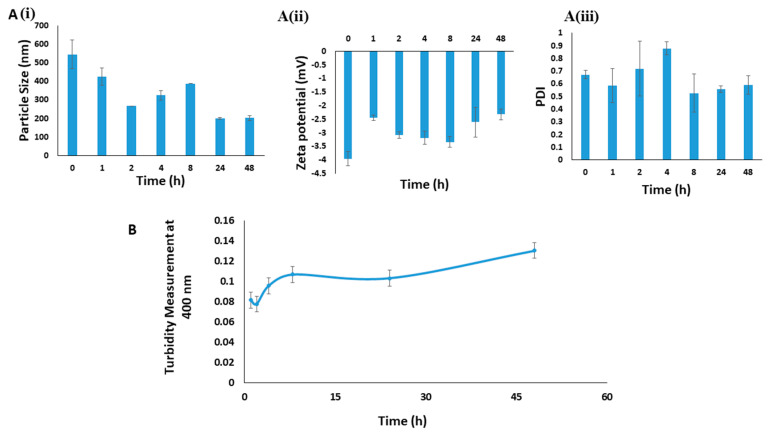
Systemic and physical stability analysis based on **A** (**i**) particle size, **A** (**ii**) PDI and **A** (**iii**) Zeta potential of Blank NaCNs for two days at 37 °C (**B**) Turbidity measurement of NaCNs for two days at 37 °C.

**Figure 12 pharmaceutics-12-00984-f012:**
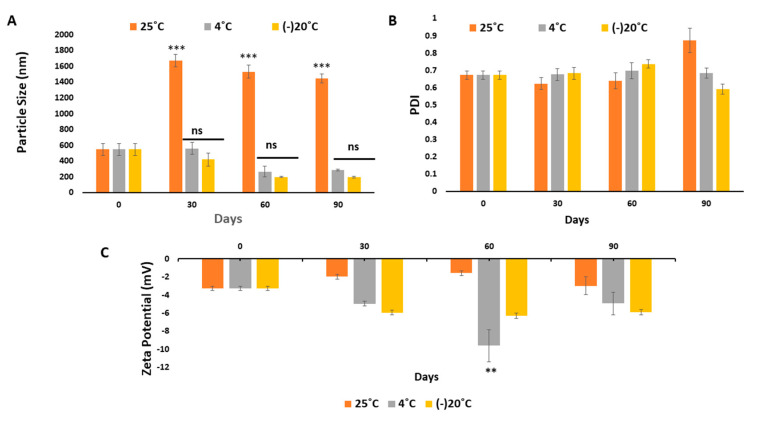
Physical stability analysis based on (**A**) particle Size, (**B**) PDI and (**C**) Zeta potential of Blank NaCNs for three months when placed at 25, 4 and −20 °C. The values were found to be very significant (***) (*p* < 0.001) at CI of 95% vs. particle size measured at 4 and −20 °C, whereas values were significant (**) (*p* < 0.01) at CI of 95% vs. zetapotential measured at 4 and −20 °C.

**Figure 13 pharmaceutics-12-00984-f013:**
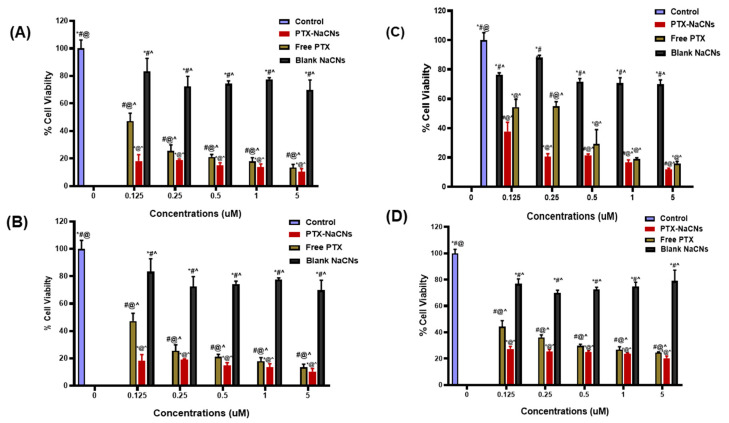
In vitro cell viability analysis through MTT assay following exposure of free PTX, PTX-NaCNs and blank micelles to (**A**) MCF-7 24 h (**B**) MCF-7 48 h (**C**) MDA-MB 231 24 h and (**D**) MDA-MB 231 48 h cell lines at 0.125–5.000 µM after 24 h and 48 h incubation. Data are shown as mean ± S.D. where *n* ≥ 3 and ^ *p* < 0.0001 vs. control (media), ^@^
*p* < 0.0001 vs. Blank micelles, * *p* < 0.0001 vs. Free PTX, # *p* < 0.0001 vs. PTX-NaCNs.

**Figure 14 pharmaceutics-12-00984-f014:**
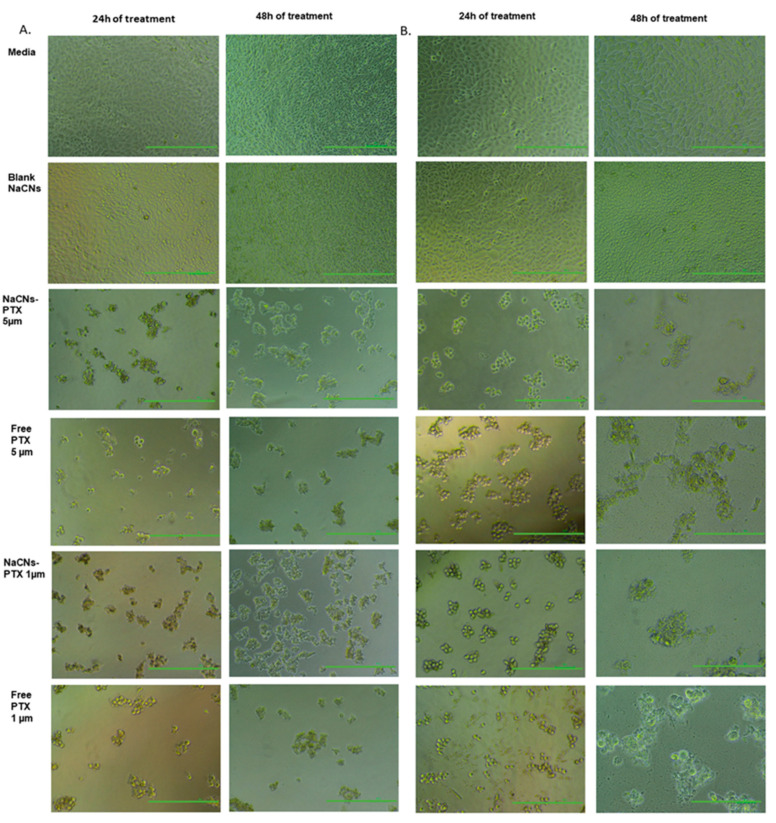
Optical images of (**A**) MCF-7 cells and (**B**) MDA-MB 231 cell lines, treated with control (media), Blank NaCNS, Free PTX and PTX-NaCNs at 1 and 5 µM (Scale bar: 100 μm).

**Figure 15 pharmaceutics-12-00984-f015:**
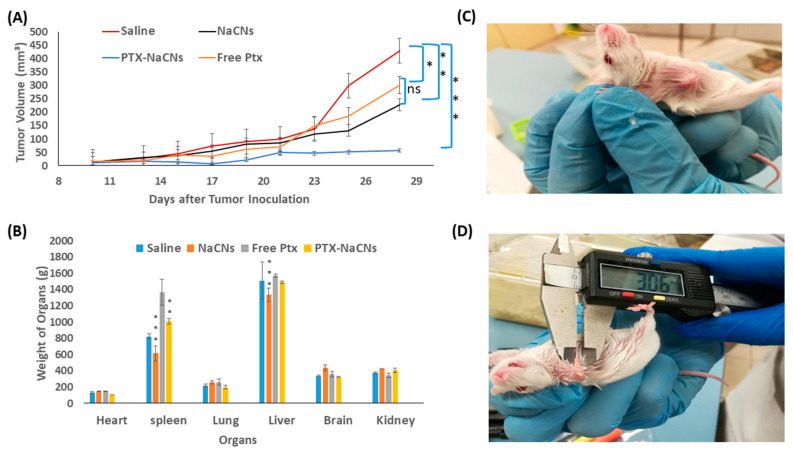
In vivo anti-tumor efficacy showing (**A**) mean tumor volume (mm^3^) of negative control, NaCNS, free PTX and PTX-NaCNs-treated groups throughout the study period following tumor inoculation and (**B**) the mean weight of the vital organs of negative control, NaCNs-, free PTX and PTX-NaCNs-treated groups on day 28 after the mice were sacrificed via cervical dislocation. (**C**) Murine breast cancer model selected to evaluate the anti-tumor efficacy of PTX-NaCNs. (**D**) Tumor measurement using a digital Vernier calliper [values are significant (*) at *p* < 0.05, very significant (**) at *p* < 0.01 and highly significant (***) at *p* < 0.001].

**Figure 16 pharmaceutics-12-00984-f016:**
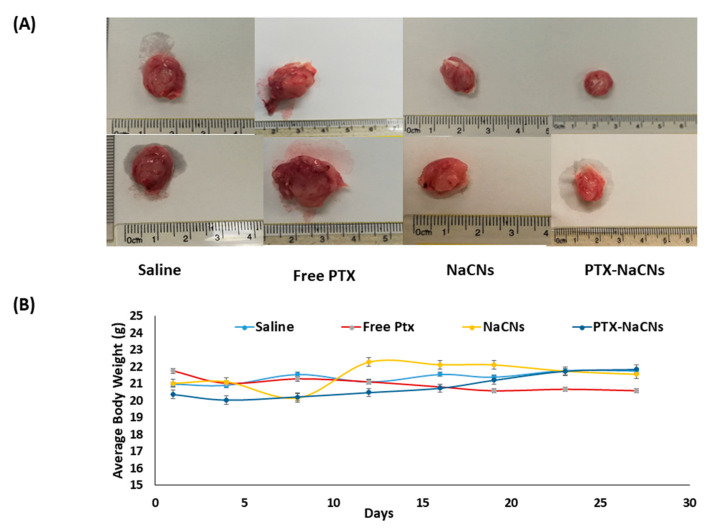
(**A**) Excised tumor images illustrating the different anti-tumor efficacy between the groups. (**B**) Mean body weight curves of tumor-induced mice treated with NaCNs and PTX-NaCNs compared with the free PTX and saline-treated groups over the treatment period of 28 days.

**Table 1 pharmaceutics-12-00984-t001:** Characteristic of Blank and PTX-NaCNs.

Characterization	Blank NaCNs	PTX-NaCNs
Particle Size (nm)	332.700 ± 48.640	198.400 ± 9.810
PDI	0.664 ± 0.064	0.469 ± 0.054
Zeta potential (mV)	−1.870 ± 0.350	−1.860 ± 0.278
DL (% *w*/*w*)	-	3.569 ± 0.460
E.E. (% *v*/*v*)	-	50.982 ± 4.000

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
