# Peer review of "Optimization and Formulation of Nanostructured and Self-Assembled Caseinate Micelles for Enhanced Cytotoxic Effects of Paclitaxel on Breast Cancer Cells"

_pharmaceutics, 2020, doi:10.3390/pharmaceutics12100984_

Round 1

Reviewer 1 Report

This is an excellently written manuscript by an outstanding team of authors that have published quite well in the past on the same subject. This paper talks about the use of NaCNs as drug delivery agents for the delivery of PTX to breast cancer cells. This article first shows extensive characterization of the NaCN particles via multiple means and then goes on to perform in vitro and in vivo studies on the same. This was an article that is so well written that the authors address a lot of concerns I was having when reading the article then and there. It was easy to follow and seems very professionally written. An outstanding article overall to read for anyone in the field who is planning to learn more about this process. 

There are some minor concerns that need to be addressed.

  1. A significance test needs to be performed for figures 1a-d. A simple T-test should suffice
  2. Some sort of quantification for figure 4 might add some value to the reader to understand what the author is trying address.
  3. SImilar quantification and stats on figure 5 will be beneficial
  4. Figure 8, the scale bars are too small to see. they may need to be re-drawn and labelled.
  5. For Figure 9, some information regarding what is the composition usually for sodium caseinate is necessary. In fact the EDS analysis may barely provide any important information and hence may even be removed from the main figure list.
  6. Some references to previous papers that confirm the characteristic NaC FTIR spectrum is necessary if the authors wish to confirm their findings.
  7. Figure 12, it would be beneficial to address why the particle size increased for the micelles kept at room temp. Also the authors should perform T-test on 12c
  8. If the authors can show some cytotox data on primary cells also, it will add significant importance to the data shown. While the authors already have in vivo data, this is not necessary, but would recommend.
  9. The scale bars for figure 14 are not visible.
  10. In Figure 16, if the authors can show more than 1 tumor sample per group, it will definitely increase the confidence of their study.

Overall, this is an excellent article. The authors have previously published similar dataset for doxorubicin and hence the novelty of the data is a little low. Nevertheless a very important article.

Author Response

Reviewer: 1

Comments to the Author

This is an excellently written manuscript by an outstanding team of authors that have published quite well in the past on the same subject. This paper talks about the use of NaCNs as drug delivery agents for the delivery of PTX to breast cancer cells. This article first shows extensive characterization of the NaCN particles via multiple means and then goes on to perform in vitro and in vivo studies on the same. This was an article that is so well written that the authors address a lot of concerns I was having when reading the article then and there. It was easy to follow and seems very professionally written. An outstanding article overall to read for anyone in the field who is planning to learn more about this process. 

There are some minor concerns that need to be addressed.

  1. Significance test needs to be performed for figures 1a-d. A simple T-test should suffice

Response: Two way analysis was carried out for statistical analysis as suggested by the reviewer.

  1. Some sort of quantification for figure 4 might add some value to the reader to understand what the author is trying address

Response: More description in the results and discussion section has been added for further explanation to this experiment (line no 394-396).

“Thus the data generated through the optical microscope further validated our hypothesis that 1mg/mL of sodium caseinate is the suitable concentration to prepare micelle within one hour of mixing time.”

However, there is no software or technique currently available to quantitate.

  1. Similar quantification and stats on figure 5 will be beneficial.

Response: More description in the results and discussion section has been added for further explanation to this experiment (line no 407-409).

“Thus more detailed nanostructured analysis through FESEM further proved that 1 mg/mL concenteration of sodium casienate is deemed suitable to prepare micelle in one hour of mixing time”.

However, there is no software or technique to quantitate.

  1. Figure 8, the scale bars are too small to see. They may need to be re-drawn and labelled.

Response: Scale bar has been added to figure 8.

  1. For Figure 9, some information regarding what is the composition usually for sodium caseinate is necessary. In fact the EDS analysis may barely provide any important information and hence may even be removed from the main figure list.

Response: We have added the concentration of NaCNs, and transferred the figure to supplementary figure (S1).

  1. Some references to previous papers that confirm the characteristic NaC FTIR spectrum is necessary if the authors wish to confirm their findings.

Response: These characteristics peak were also observed in previous studies [53, 54] and mentioned references have been included in the manuscript (line no 501).

  1. Figure 12, it would be beneficial to address why the particle size increased for the micelles kept at room temp. Also the authors should perform T-test on 12c

Response: The reason for the increase in particle size of the micelles kept at room temp has been added to the manuscript (line no 565-567).

“At higher temperature, the increased thermal energy might induce the aggregation of micelles by accelerating the interactions between them; hence at 250C the particle size increased.”

T-test has been performed on fig 13.

  1. If the authors can show some cytotoxic data on primary cells also, it will add significant importance to the data shown. While the authors already have in vivo data, this is not necessary, but would recommend.

Response: The ultimate goal is to deliver the drugs to the tumor with the help of nano micelles through passive targeting harnessing the EPR effect while minimizing the off-target distribution to other organs, so that normal healthy cells are less affected.

  1. The scale bars for figure 14 are not visible.

Response: Scale bars of figure 14 (changed to fig 15) have been corrected to make them visible.

  1. In Figure 16, if the authors can show more than 1 tumor sample per group, it will definitely increase the confidence of their study.

Response: 1 more tumor sample has been added to figure 17 B as per suggestion (line no 689-691).

Reviewer 2 Report

  1. 2.11. In vivo study: Authors used Balb/c mice to induce breast tumor model, not using nude mice. Hows the success rate of induced process?
  2. Figure 1: zeta size is informal. Particle size or size can have called.
  3. Figure 2 showed the raw data, authors can present more smart.
  4. Figure 2 presented the diversity zeta potential value at different mixing time. Authors may explain why diversity zeta potential value.
  5. Figure 7A showed the large broad distribution of the size. Authors need explain.
  6. Figure 13: In vitro cell viability analysis showed that the treatment of PTX-NaCNs for 48, the cell viability was higher. Authors need explain. Moreover, the Scale labeled can be improved.
  7. Figure 12 showed the blank formulation. However, the drug is compatible in the formulations are more importantly. Authors need explain.
  8. Figure: 16B showed * in the figure, authors require correct.
  9. Authors can explain why screen MCF-7 and MDA-MB 231, and what the different outcome.Moreover, 4T1 cells were TNBC like, it's totally different with MCF-7 cells.
  10. the figures require rearrangement.

Author Response

Reviewer: 2
1. 2.11. In vivo study: Authors used Balb/c mice to induce breast tumor model, not using nude mice. How’s the success rate of induced process?

Response: Mouse 4T1 mammary cancer cells were used to induce tumors in the mammary pad of Balb/c mice. The success rate of the induced process was more than 96 %.

2. Figure 1: “zeta size” is informal. Particle size or size can have called.

Response: In Fig. 1, the word ‘zeta size’ has been changed to ‘particle size’.

  1. Figure 2 showed the raw data, authors can present more smart.

Response:  Figure 2 has been redrawn to present the data more nicely.

  1. Figure 2 presented the diversity zeta potential value at different mixing time. Authors may explain why diversity zeta potential value.

Response: The above concern has been addressed in the manuscript (line no 350-352) and mentioned below.

“Moreover, due to different electrostatic interactions between casein molecules, affecting casein stability, we obtained diverse zeta potential values in the samples prepared with different casein concentrations and mixing time.”

  1. Figure 7A showed the large broad distribution of the size. Authors need explain.

Response: Broad size distribution occurred due to the nature of casein reassembly, but as the figure shows, more than 90 percentage of micelle lies in the ⁓ 200 nm, indicating the suitable particle size of our micelle (449-452).

  1. Figure 13: In vitro cell viability analysis showed that the treatment of PTX-NaCNs for 48, the cell viability was higher. Authors need explain. Moreover, the Scale labeled can be improved.

Response: The above concern has been addressed in the manuscript (line no 601-603) and mentioned below.

“PTX-NaCNs tend to be potent and showed immediate effect after 24 h, however due to nontoxicity of the nanomicelles some of the cells might further proliferate, showing slightly higher cell viability at 48 h as depicted in figure 14, B&D”.

The scale bar of the figure 13 (now fig 14) has been changed.

7. Figure 12 showed the blank formulation. However, the drug is compatible in the formulations are more importantly. Authors need explain.

Response: The above concern has been addressed (line no 572-574) and mentioned below.

The need of the drug compatibility inside the formulations was deemed necessary. However, here we optimized the micelle to observe the micelle stability without drug loading over a period of time”

8. Figure: 16B showed * in the figure, authors require correct.

Response:  The mentioned typo error has been corrected.

  1. Authors can explain why screen MCF-7 and MDA-MB 231, and what the different outcome. Moreover, 4T1 cells were TNBC like, it's totally different with MCF-7 cells. 

Response: Both concerns have been addressed in the manuscript and mentioned below

“Most of the formulations demonstrated excellent results in MCF-7 but found to be resistant in MDA-MB 231 cells due to their resistant nature”.

“4T1 cell lines were used to induce tumor in mice. Although 4T1 cells are quite different from the MCF-7 cells, as triple negative breast cancer cells (TNBC) they have similar resistance nature as MDA-MB 231 where PTX-NaCNs showed promising cytotoxic results.” (line no 590-591, 662-664).

10. The figures require rearrangement.

Response: Figures were rearranged as per the reviewer’s request.

Reviewer 3 Report

In the present study, the authors developed self-assembled sodium casinate nanomicelles (NaCNs) with paclitaxel (PTX) to improve the efficacy and safety of PTX for cancer treatment, and successfully achieved to improve the solubility, cell uptake efficiency and antitumor activity of PTX. On the other hand, there seems to be room for improvement in drug release and formulation stability from PTX-NACNs.

  1. Table 1: The PDI of blank and PTX-NaCNs was higher than 0.3. I think these particles are not fully uniform. Did the authors optimize the preparation method? Please explain.

  1. Figure 11B and 13: The cellular uptake of PTX-NaCNs was four times higher than that of free PTX (Fig. 11), but the cytotoxic effect of PTX-NaCNs is almost the same as that of PTX (Fig. 13). Please discuss it.

  1. Abstract and Fig. 12A, C: The authors described ”the nanomicelles also presented improved colloidal stability for three months (line 37)”. On the other hand, Fig. 12C showed the gradual decrease of particle size at 4°C or -20°C. The authors should show the physicochemical characteristics (particle size, zeta potential and PDI) of particles after three months.

  1. Figure 11 and 15: The authors selected the different cell lines in in vitro and in vivo experiments. Since the effect of antitumor reagent depends on the cell types, the authors should evaluate the effect using the same cell type or explain the reason why the authors selected the cell lines.

  1. Figure 13 and 15: The authors should discuss the toxicity of blank NaCNs. Figure 15B showed that blank NaCNs group decreased the weight of the spleen and liver. Did the authors evaluate the cytotoxicity?

  1. Line 699:The authors described ”Free PTX-treated groups presented a significant (p<0.001) increase in the spleen and liver size”. However, there was no significant difference in liver size between the saline group and Free PTX-treated group (Figure 15B). Please correct.

  1. Figure 15A:The authors need to evaluate the pharmacokinetics of PTX-NaCNs. Did PTX-NaCNs deliver to tumor through EPR effect?

  1. Please correct any typos; Scale bar: 100 μM (Lane 408) and spleen weight (Figure 15B), etc.

Author Response

Reviewer 3

Comments to Authors

In the present study, the authors developed self-assembled sodium casinate nanomicelles (NaCNs) with paclitaxel (PTX) to improve the efficacy and safety of PTX for cancer treatment, and successfully achieved to improve the solubility, cell uptake efficiency and antitumor activity of PTX. On the other hand, there seems to be room for improvement in drug release and formulation stability from PTX-NACNs.

Response: Really thankful for your valuable comments.

  1. Table 1: The PDI of blank and PTX-NaCNs was higher than 0.3. I think these particles are not fully uniform. Did the authors optimize the preparation method? Please explain.

Response:  Preparation method was optimized through various means by changing the casein concentration, changing the mixing speed and also changing the mixing time. Furthermore the explanation to heterogeneous particles has been added in the manuscript (line no 449-457).

  1. Figure 11B and 13: The cellular uptake of PTX-NaCNs was four times higher than that of free PTX (Fig. 11), but the cytotoxic effect of PTX-NaCNs is almost the same as that of PTX (Fig. 13). Please discuss it.

Response: The above concern has been addressed in the manuscript (line no 614-617) and mentioned below.

“Additionally, compared to three controls (untreated, blank micelles and free PTX), PTX-loaded micelles showed considerably higher cytotoxicity due to efficient binding of PTX to the micelles, and the results were further correlated with our in-vivo studies, where PTX-loaded micelles significantly lowered the size of tumor as compared to free PTX and the blank micelles.”

  1. Abstract and Fig. 12A, C: The authors described ”the nanomicelles also presented improved colloidal stability for three months (line 37)”. On the other hand, Fig. 12C showed the gradual decrease of particle size at 4°C or -20°C. The authors should show the physicochemical characteristics (particle size, zeta potential and PDI) of particles after three months.

Response:  At 4°C and -20°C, there was a slight decrease in the particle size (⁓ 200 nm ) apparently; however, the change was found to be non-significant after t-test.  The particles with average size of ⁓ 200 nm  are highly expected to be efficien­tly taken up by tumor cells due to EPR effect.

As mentioned in the manuscript (line no 565-567): “At higher temperature, the increased thermal energy might induce the aggregation of micelles by accelerating the interactions between them; hence at 250C the particle size increased.”

  1. Figure 13 and 15: The authors should discuss the toxicity of blank NaCNs. Figure 15B showed that blank NaCNs group decreased the weight of the spleen and liver. Did the authors evaluate the cytotoxicity?

Response: Blank NaCNs demonstrated more than 75% cell viability (Figure 13) in almost all concentrations of sodium caseinate used to prepare NaCNs as reported in previous research and the reference has been added in the manuscript [67]. The spleen size of normal mouse is almost equivalent to the spleen size found in NaCNs-treated group, indicating NaCNs nontoxicity, and due to the non-toxic nature of the micelle, the sizes of both organs (spleen and liver) were observed significantly bigger, compared to free PTX group. Furthermore, no mice died or lose weight in NaCNs-treated group, indicating nontoxicity of NaCNs.

  1. Line 699:The authors described ”Free PTX-treated groups presented a significant (p<0.001) increase in the spleen and liver size”. However, there was no significant difference in liver size between the saline group and Free PTX-treated group (Figure 15B). Please correct.

Response: The statistical difference of free PTX was compared to the PTX-NaCNs, as mentioned in the manuscript (line no 701-702).

  1. Figure 15A:The authors need to evaluate the pharmacokinetics of PTX-NaCNs. Did PTX-NaCNs deliver to tumor through EPR effect?

Response: We appreciate your suggestion. However, since a new animal ethics approval is required to conduct the pharmacokinetic study, we are afraid that it would take at least several months to get the relevant data.

As mentioned in the manuscript (Line no 689-690), NaCNs facilitated the passive delivery and retention PTX into tumor cells via an EPR effect [23] with enhanced anti-tumor efficacy.

  1. Please correct any typos; Scale bar: 100 μM (Lane 408) and spleen weight (Figure 15B), etc.

Response: The typo error has been corrected.

Round 2

Reviewer 2 Report

  1. Figure 1. “Zeta size” still present in the figure 1. Moreover, The statistic mark should be improve.
  2. Figure 2. The units of zeta potential should be addressed. Moreover, authors require do the statistics.
  3. Figure 10 required improved.  Figure 10 (A): The error bar and horizontal may give some improve. it makes reader confuse.

Author Response

Reviewer: 2

Comments to Authors

1. Figure 1. “Zeta size” still present in the figure 1. Moreover, The statistic mark should be improve.

Response: The word Zeta size has been removed from the figure 1 and replaced by the particle size and statistics marks have been improved further.

2. Figure 2. The units of zeta potential should be addressed. Moreover, authors require do the statistics.

Response: The unit of zeta potential has been added in figure 2 and further statistical analysis has been performed.

3. Figure 10 required improved.  Figure 10 (A) The error bar and horizontal may give some improve. it makes reader confuse.

Response: The Figure 10 has been improved further as suggested by the Reviewer.

Reviewer 3 Report

The authors have adequately answered all my concerns and I am satisfied with the revised manuscript.

Author Response

Reviewer: 3

Comments to Authors

The authors have adequately answered all my concerns and I am satisfied with the revised manuscript,

Response: Many thanks for your comments.